# Clinical Behavior of Ceramic, Hybrid and Composite Onlays. A Systematic Review and Meta-Analysis

**DOI:** 10.3390/ijerph17207582

**Published:** 2020-10-19

**Authors:** Naia Bustamante-Hernández, Jose María Montiel-Company, Carlos Bellot-Arcís, José Félix Mañes-Ferrer, María Fernanda Solá-Ruíz, Rubén Agustín-Panadero, Lucía Fernández-Estevan

**Affiliations:** Department of Dental Medicine, Faculty of Medicine and Dentistry, University of Valencia, 46010 Valencia, Spain; naiabustamante@gmail.com (N.B.-H.); jose.maria.montiel@uv.es (J.M.M.-C.); Carlos.Bellot@uv.es (C.B.-A.); jmanes@uv.es (J.F.M.-F.); m.fernanda.sola@uv.es (M.F.S.-R.); lucia.fernandez-estevan@uv.es (L.F.-E.)

**Keywords:** posterior partial restorations, onlays, ceramics, hybrid dental material, composite, clinical evaluation, survival rate, complications

## Abstract

A systematic review and meta-analysis was performed to analyze the survival of onlay restorations in the posterior region, their clinical behavior according to the material used (ceramic reinforced with lithium disilicate, conventional feldspathic ceramic or reinforced with leucite; hybrid materials and composite), possible complications, and the factors influencing restoration success. The systematic review was based on the preferred reporting items for systematic reviews and meta-analyses (PRISMA) statement, without publication date or language restrictions. An electronic search was made in the PubMed, Scopus, Embase, and Cochrane databases. After discarding duplicate publications and studies that failed to meet the inclusion criteria, the articles were selected based on the population, intervention, comparison, outcome (PICO) question. The following variables were considered in the qualitative and quantitative analyses: restoration survival rate (determined by several clinical parameters), the influence of the material used upon the clinical behavior of the restorations, and the complications recorded over follow-up. A total of 29 articles were selected for the qualitative analysis and 27 for the quantitative analysis. The estimated restoration survival rate was 94.2%. The predictors of survival were the duration of follow-up (beta = −0.001; *p* = 0.001) and the onlay material used (beta = −0.064; *p* = 0.028). Composite onlays were associated with a lower survival rate over time. Onlays are a good, conservative, and predictable option for restoring dental defects in the posterior region, with a survival rate of over 90%. The survival rate decreases over time and with the use of composite as onlay material.

## 1. Introduction

Dental structural defects can be a consequence of a range of factors, though caries are the predominant cause, with an estimated prevalence of over 90% in the worldwide general population [1,2,3]. Other factors such as erosion, abrasion, wear, fracture, and their combinations may contribute to early hard dental tissue loss. These situations can give rise to sensitivity, pain, pulp tissue involvement, secondary caries, and periodontal and occlusal problems (antagonistic extrusion, interferences, etc.), among other disorders, resulting in a need for restoration [4,5,6,7,8]. Partial restorations are an alternative to conventional crowns, in view of the growing demand for minimally invasive restorations, since crowns or complete covering restorations imply an important loss of tooth structure, with macromechanical and more invasive preparation of the dental tissues [7,9,10,11,12,13]. In this regard, partial restorations have become a conservative treatment option thanks to their good aesthetic outcomes, durability, color stability, biocompatibility and high long-term survival rates [14,15,16]. The development of this treatment option has allowed lesser tooth reduction during preparation of the restoration, and thus greater preservation of the dental tissues. Partial restorations (Figure 1) are classified according to the area to be restored as inlays (without covering the cusps), onlays (covering at least one cusp) [17], and overlays (covering all cusps) [18,19,20,21].

In addition to classification based on the area to be restored, two types of partial restorations have been defined according to the manufacturing method employed: direct and indirect restorations. In direct restorations, the restoring material (which is limited to composite) is placed directly in the defect or cavity, affording greater preservation of tooth structure. These restorations are mainly indicated in cases of lesser dental destruction [22,23,24,25,26]. The indirect restoration technique in turn involves the preparation of the restoration outside the mouth (using composite, hybrids, or ceramics) (Figure 2).

Compared with the direct technique, the indirect restoration approach affords cusp protection, with reinforcement of the compromised tooth [19,21,27,28]. within addition to the evolution of adhesive techniques and materials, modern digital methods for manufacturing indirect restorations have been developed. Computer-aided design and computer-aided manufacturing (CAD-CAM) systems, along with the improvement of intraoral scanners, constitute alternative to the conventional method for manufacturing high-quality indirect restorations [29,30]. These novel approaches allow the restoration of lost tooth tissues in a single visit, with consequent reduction of the overall treatment time [31]. Nevertheless, the limitations of the different techniques (analog and digital) must be known in order to establish the workflow best suited to each case [32], since correct manipulation of the material and adequate selection of the manufacturing or adhesion technique are key factors influencing restoration success or failure [23]. The use of such restorations has increased as a result of advances in adhesion and cementing technologies, affording greater bonding between the tooth and the restoration material, and thus allowing for more conservative [33,34,35,36]. Restoration materials have also been influenced by these advances, evolving from the use of materials such as gold or amalgam, with a long history of clinical success and biocompatibility [23], to more current materials such as ceramics (conventional feldspathic ceramic, leucite-reinforced ceramic, lithium disilicate ceramic), hybrid materials (resin nanoceramic and hybrid ceramic) or composite resins [37,38]. These materials have different chemical compositions that explain most of their clinical properties. Ceramic materials are fragile and more vulnerable to fracture than composite materials, though they are also harder than the latter and are therefore more resistant to wear. However, for this same reason they may induce wear of the surface of the opposing tooth [1,39]. Hybrid materials in turn share characteristics common to both ceramics and composite materials, with an elastic modulus similar to that of the natural tooth. Furthermore, in the same way as composite materials, they are easy to adjust, repair, or modify [37].

Partial restorations are not without complications, however, including fractures, possible tooth hypersensitivity, adjustment problems and marginal integrity loss, microleakage and adhesion failure. Other factors that affect the clinical performance of such restorations are material wear or wear of the opposing teeth, plaque accumulation, gingivitis, secondary caries, and instability of color or anatomical shape or radiopacity [40,41]. The differences in the mechanical properties of ceramics and resin-based materials are very important in determining possible complications or failure of the restorations.

The present systematic review and meta-analysis was carried out to analyze the survival of onlay restorations in the posterior region, examining the different materials used and identifying the types and frequencies of complications reported in randomized controlled trials (RCTs) and retrospective and prospective studies.

Two types of partial restorations have been defined, according to the technique employed: direct restorations and indirect restorations. In the former case the restoration material is placed directly in the defect or cavity, affording greater preservation of tooth structure. These restorations are mainly indicated in cases of lesser dental destruction [22,23,24,25,26]. The indirect restoration technique in turn involves preparation of the restoration outside the mouth, based on an imprint impression or model of the tooth. This procedure affords improved physical and mechanical properties on subsequently polymerizing the material with light or heat. It also offers ideal occlusal morphology, wear compatible with that of the opposing natural dentition, and avoids the polymerization contraction found with direct partial restorations. The indirect technique is indicated for the reconstruction of class II cavities with major destruction in the interdental (interproximal) zone; extensive isthmuses measuring over a third of the width of the occlusal surface; and the reconstruction of one or more cusps. However, the indirect restoration technique is more time consuming, and is associated with added costs and a greater number of patient visits [23,26].

Onlay survival or longevity is conditioned by a range of factors including the condition of the supporting teeth, patient habits or clinical protocols, and the properties of the restoration material used [1,37]. The development of new materials has allowed lesser tooth reduction during preparation of the restoration, and thus greater preservation of the dental tissues [37]. These materials have different chemical compositions that explain most of their clinical properties. In this regard, ceramic materials comprise ceramics with a vitreous or vitroceramic matrix, including conventional feldspathic ceramic, synthetic ceramics (leucite-reinforced and/or lithium disilicate ceramic) and hybrid materials or ceramics with a resin matrix [37,38]. In turn, composites are used for the preparation of indirect restorations [21,37,38,42].

However, studies on the long-term behavior of these materials are lacking.

## 2. Materials and Methods

A systematic literature review was made, based on the preferred reporting items for systematic reviews and meta-analyses (PRISMA) statement (http://www.prisma-statement.org) and the Cochrane Manual for the conduction of systematic reviews and meta-analyses. The study was registered in the PROSPERO database (Ref. CRD42019126755).

The population, intervention, comparison, outcome (PICO) question was: What are the clinical behavior and the possible complications of posterior region onlays according to the material used for the restoration? Specifically, “P” (population) refers to the type of patients studied, i.e., individuals subjected to partial restoration of the posterior region, “I” (intervention) refers to onlay restoration of the posterior region, ‘’C’’ (comparison) refers to comparison of the different restoration materials cited in the literature, and “O” (outcome) refers to the clinical behavior and possible complications over time.

The search strategy was based on combinations of the following keywords: onlay, overlay, occlusal veneer, coverage (type of restoration); dental ceramics, hybrid material, zirconia, composite or cad-cam (type of material); clinical evaluation, clinical trial, longevity, success, failure, survival rate, clinical performance, follow-up study, comparative study (clinical evaluation).

The search strategy included three MeSH terms: “clinical trial”, “survival rate”, and “follow-up study”. The boolean operators “OR” and “AND” were used.

The following search was carried out:

(“onlay*” OR “overlay*” OR coverage OR occlusal veneer) AND (“hybrid material*” OR “dental ceramics*” OR zirconia OR composite OR “CAD/CAM”) AND (“clinical evaluation” OR “clinical trial” OR “longevity” OR “success” OR “failure” OR “survival rate” OR “clinical performance” OR “follow up study” OR “clinical study” OR “comparative study”).

A thorough search was made of the United States National Library of Medicine National Institutes of Health (PubMed), Scopus, Embase and Cochrane databases. The systematic review and meta-analysis covered all the international literature published up until April 2020.

The search strategy was carried out by three investigators (N.B.H., L.F.dE., and J.F.M.) on an independent basis. The studies were selected from the titles and abstracts, considering the specified inclusion and exclusion criteria. Specifically, article selection was carried out in three stages: (1) selection according to the relevance of the title; (2) selection according to the relevance of the abstract; and (3) full-text analysis and cross-comparison against the inclusion criteria. Randomized controlled trials (RCTs) and retrospective and prospective studies were considered. Systematic reviews, literature reviews, clinical cases, case series, editorials, and in vitro studies were excluded. The included studies were referred to patients over 18 years of age treated with onlays (partial restorations covering at least one dental cusp) in the posterior region, and involving a follow-up of one year or more. There were no restrictions in terms of year of publication or language. The systematic review was carried out by one of the investigators (N.B.H.), and the subsequent meta-analysis was performed by an investigator blinded to previous processing of the articles (J.M.C.).

The variables registered in each of the studies were: author, year of publication, title and journal, sample size (*n*), duration of follow-up (months), restoration material, adhesive and cementing system, survival rate, and complications.

The complications recorded in the studies were evaluated based on the modified United States Public Health Service (modified USPHS) and California Dental Association (CDA) scales. The modified USPHS criteria were used to classify the condition of the restoration from A-D (Alpha, Bravo, Charlie, Delta), as follows: A = restoration in excellent condition referred to the considered parameter, and expected to have prolonged survival over time; B = restoration in suboptimum condition and possibly needing replacement in the future; C = failure of the restoration or of the surrounding tissues; and D = failure of the restoration referred to the considered parameter.

The CDA is a variation of the USPHS system. Both instruments are based on an ordinal scale, rating restorations as being either “acceptable” or “not acceptable”. “Success” refers to successful restorations, “survival” refers to restorations that are not intact but survive, and “failure” refers to failed restorations. Success is taken to mean that the considered parameter meets the highest standard; survival is taken to mean that, although the restoration has deteriorated, replacement is not necessary; and failure is taken to mean that the restoration needs to be replaced. [8]. The evaluated parameters comprised postoperative sensitivity, fracture, interproximal (interdental) contact, and occlusal contact, among others.

The methodological quality of the studies was analyzed using two specific scales: the Newcastle–Ottawa Quality Evaluation Scale (NOS) [43] for the evaluation of cohort studies, and the PEDro scale for the evaluation of clinical trials [44].

The NOS [43] comprises 8 items yielding a potential total score of 9. Three domains are considered: patient selection; comparability of the study groups; and results or outcome. High quality studies yield 3–4 stars in the patient selection domain; 1–2 stars in the comparability domain; and 2–3 stars in the results or outcome domain. In turn, the PEDro [44] comprises 11 items scored from 0–11, according to whether the evaluated item is present or absent. Studies yielding a score of ≥ 5 are considered to be of high quality and with a low risk of bias.

For the meta-analysis, the included studies were combined by means of a random effects model. The effect size was the events rate, with calculation of the corresponding 95% confidence interval (95%CI). The statistical heterogeneity between studies was assessed based on the Q-test and I2 statistic. The presence of differences between subgroups was evaluated using the between-group Q-test. Meta-regression analysis was performed based on a mixed effects model, determining the existence of significant covariables with the moderators test. Publication bias in turn was assessed using the trim and fill method. A graphic representation of the meta-analysis was in the form of forest plots, with meta-regression being depicted in the form of scatter plots and publication bias as funnel plots. Statistical significance was considered for *p* < 0.05. The data were analyzed using the R statistical package.

## 3. Results

The initial search yielded a total of 6532 studies: 1348 identified in PubMed, 2102 in Scopus, 2865 in Embase and 217 in the Cochrane database. A total of 1959 of these articles were duplicates and were eliminated. After reading of the title and abstract, 4511 publications were excluded because they failed to meet the inclusion criteria (many were in vitro studies, and many others made no distinction between results referred to inlays and onlays). Sixty-three articles were subjected to full text evaluation. Of these, 34 were rejected because they addressed a different question; made no distinction between results referred to inlays and onlays; or were the same study published prior to another study but involving a smaller sample. A final total of 29 articles met the inclusion criteria and were subjected to qualitative analysis, and 27 were likewise subjected to quantitative synthesis or meta-analysis, since all the necessary data and variables were present (Figure 3). Of the 29 studies included in the systematic review, 12 were cohort studies and the remaining 17 were clinical trials.

The results of methodological quality assessment based on the Newcastle–Ottawa and PEDro scales are reported in Table 1 and Table 2. Most of the cohort studies were of high quality according to the Newcastle–Ottawa Scale, with a score of >6 [43] (Table 1). Only five of the 12 studies had a score of <7. On the other hand, based on the PEDro scale, six articles presented scores of >5, indicating high methodological quality. The other 11 articles yielded scores of ≤5 (Table 2). Quality was most often adversely affected because of failure to fulfill items related to subject or measurement blinding.

A total of 29 articles were entered in the qualitative analysis. The sample sizes ranged from 14–231 restorations, and the duration of follow-up varied between 24–180 months. The materials analyzed were: feldspathic ceramic reinforced with lithium disilicate, conventional feldspathic ceramic or feldspathic ceramic reinforced with leucite, hybrid materials and composite. Of the 29 articles analyzed, five of them evaluated and compared two different materials within the same research. In five articles only ceramic reinforced with lithium disilicate was analyzed, and in another three publications ceramic reinforced with lithium disilicate was compared with another material (overall, ceramic reinforced with lithium disilicate was used in eight articles). In 13 articles, only feldspathic ceramics (conventional or reinforced with leucite) were analyzed, and in another five publications feldspathic ceramics were compared with another material (overall, 18 articles used this material). Hybrid materials were analyzed in one article and in two others these materials were compared against another material (a total of three articles used hybrid materials). Composites were used in five investigations.

Eight articles analyzed feldspathic ceramic reinforced with lithium disilicate, while feldspathic ceramic (conventional or reinforced with leucite) was evaluated as restoration material in 18 articles. Two of these studies compared both materials (ceramic reinforced with lithium disilicate and feldspathic ceramic reinforced with leucite). One of them compared two lithium disilicate ceramics, differentiating their laboratory manufacture (press/CAD), and another compared two different ceramics (conventional and reinforced with leucite). In addition, we analyzed three articles in which hybrid materials were used. This type of material was compared against ceramic reinforced with leucite in one article and versus ceramic reinforced with lithium disilicate in another. In five articles the restoration material considered was composite.

The quantitative analysis combined data from 27 studies analyzing different materials such as feldspathic ceramic reinforced with lithium disilicate, conventional feldspathic ceramic or feldspathic ceramic reinforced with leucite, hybrid materials and composite.

In order to more easily distinguish the materials, feldspathic ceramic reinforced with lithium disilicate was cited as lithium disilicate or disilicate, while the other two ceramics (conventional and reinforced with leucite) were grouped as feldspathic ceramic following the nomenclature used in most of the literature consulted in the present review. The hybrid materials included products such as VitaEnamic, Cerasmart, and Lava Ultimate. The list was completed by the composite materials.

### 3.1. Percentage Survival of the Restorations

The duration of follow-up ranged from 24–180 months. A random effects model estimated a percentage survival of 94.2% (95%CI 92.3–96.1), with a prediction interval of between 84.0% and 100% (Figure 4). The observed heterogeneity between studies (Q-test = 220.8; *p* < 0.001) was considered to be high (I^2^ = 84.1%).

In the analysis of restoration material subgroups (Table 3), to explain the observed heterogeneity, on combining the data with a random effects model, we recorded statistically significant differences attributable to the restoration material used (between-groups Q-test = 13.7; *p* = 0.003). Composite materials showed a lower percentage survival (90%), while hybrid and disilicate materials yielded higher percentage survivals (99% and 98%, respectively).

Meta-regression analysis (Figure 5) with the mixed effects model (test of moderators = 16.9; *p* < 0.001; R2 or proportion of the variance explained = 16.3%), including the restoration material and duration of follow-up as moderators, showed the duration of follow-up and composite material to be significant moderators in the estimated model (Table 4) with regard to the percentage survival of the restoration. The prediction interval has been included.

The graphic representation (Figure 6) of the percentage survival predictive model was generated from the equation of the straight line, 1.03 − (0.001 × month of follow-up) − (0.07 × composite).

### 3.2. Reasons for Restoration Failure

With regard to the reasons for restoration failure, we included a total of 8 studies involving three materials: feldspathic ceramic reinforced with disilicate, feldspathic ceramic (conventional or reinforced with leucite) and composite in a combined subgroups analysis with a random effects model. There were no statistically significant differences referred to the restoration material used (between-groups Q-test = 0.55; *p* = 0.758) or to the reason for failure (between-groups Q-test = 9.05; *p* = 0.249). Composite material was associated to a 1% failure rate. Fracture was the most important reason for restoration failure (4%), followed by discoloration (1%).

### 3.3. Clinical Evaluation of the Restorations Using the Modified USPHS Criteria

We examined 12 studies that analyzed a series of clinical parameters of the restorations classified into four categories according to the clinical evaluation (Alpha, Bravo, Charlie, Delta) (Figure 7)—with Alpha representing the best evaluation for the studied parameters and Delta the poorest. These clinical parameters were combined with a random effects model and analyzed by subgroups.

A total of 89.8% of the restorations were classified as corresponding to category Alpha (95%CI 87.5–92.1). Differences were observed between the studied clinical parameters (Q-test = 61.48; *p* < 0.0001). Accordingly, a greater percentage of restorations corresponded to category Alpha on assessing the parameter marginal fracture (100%) and body fracture (99%), while a lower percentage of restorations corresponded to category Alpha on assessing the parameters surface texture (80%) or color (84%).

A total of 9.8% of the restorations were classified as corresponding to category Bravo (95%CI 7.7–11.9). Differences were observed between the studied clinical parameters (Q-test = 59.27; *p* < 0.0001). Accordingly, a greater percentage of restorations corresponded to category Bravo on assessing the parameter surface texture (21.7%) and color (16.2%), while a lower percentage of restorations corresponded to category Bravo on assessing parameters such as retention or secondary caries (0%).

A total of 0.1% (95%CI 0–0.3) and 0% (95%CI 0–0.002) of the restorations were classified as corresponding to categories Charlie and Delta respectively. No differences were observed between the studied clinical parameters (Q-test = 4.74; *p* = 0.855 and Q-test = 0.12; *p* = 1), reflecting that practically none of the restorations corresponded to categories Charlie or Delta for any of the evaluated clinical parameters.

### 3.4. Clinical Evaluation of the Restorations as Success, Survival or Failure

Twelve studies analyzing a series of clinical parameters defining the conditions of success, survival and failure were combined with a random effects model and analyzed by subgroups corresponding to the different evaluated parameters.

A total of 77.6% of the restorations (95%CI 73.6–81.8) were classified as success—indicating that the restorations met the highest quality standards (Figure 8). Significant differences were recorded between the percentages referred to the different analyzed parameters (Q-test = 79.3%; *p* < 0.0001). The percentages of success varied from the highest values corresponding to hypersensitivity (100%) and caries (98.5%) to the lowest values corresponding to marginal integrity (66.4%) and surface (50.3%).

A total of 19.4% of the restorations (95%CI 16.5–22.4) were classified as survival—indicating that the restorations suffered some type of deterioration without having to be replaced (Figure 9). Significant differences were recorded between the percentages referred to the different analyzed parameters (Q-test = 90.4%; *p* < 0.0001). The percentages of success varied from the highest values corresponding to surface (46.6%) to the lowest values corresponding to caries or hypersensitivity (0%).

A total of 0.79% of the restorations (95%CI 0.28–1.30) were classified as failure—indicating that the restorations had to be replaced (Figure 10). No significant differences were recorded between the percentages referred to the different analyzed parameters (Q-test = 6.92; *p* = 0.327). The highest percentages of failure corresponded to marginal integrity (2.32%), anatomy (2.17%), and caries (1.22%). The rest of the parameters presented values of 0%.

### 3.5. Publication Bias

An analysis was made of publication bias based on the Trim and Fill method for adjustment of the asymmetry of the funnel plot. The estimated percentage survival was 97.8% with a confidence interval of 95.6–99.9% that differed slightly from the initial estimation of 94.2% (95%CI 92.3–96.1), indicating a low probability of publication bias (Figure 11).

## 4. Discussion

The present systematic review and meta-analysis has examined onlay restorations and the restoration materials used in the literature, and has sought to identify the associated complications and their incidence. Most of the investigations and meta-analyses published to date have evaluated restorations of this kind centered on a single restoration material. Many of the published studies do not differentiate the results obtained according to the extent of the restoration (inlay or onlay). Furthermore, to the best of our knowledge, no studies have examined all the available materials for the manufacture of onlay restorations with the purpose of assessing their clinical behavior over time and of defining the gold standard or ideal material for the preparation of such restorations. In this regard, the present meta-analysis has examined the survival and possible complications of partial restorations in the posterior region, including all materials currently available for the manufacture of onlays, among which are the novel hybrid CAD-CAM materials.

The present review showed onlays in the posterior region to have a survival rate of 94.2% (95%CI 92.3–96.1), with a prediction interval of 84.0–100%. This high survival rate confirms the capacity of onlay indirect partial restorations to repair dental structural defects in the posterior region in a conservative and predictable way [23,26,36].

Statistically significant differences were observed according to the material used for the restoration (between-groups Q-test = 13.7; *p*-value = 0.003). Composite was associated to lower percentage survival (90%) than materials such as hybrids and disilicate (99% and 98%, respectively). Similar results were obtained by Mangani [26], who found ceramic restorations to offer a higher survival rate (94.9%) than composite restorations (91.1%). Likewise, similar data were obtained in different investigations in which ceramic restorations yielded an 88.7% success rate at 10 years, while composite resins presented a success rate of 84.78% at five years of follow-up [30,70]. Although some articles have concluded that there is little evidence regarding the superior performance of ceramic materials versus composites as onlay restoration materials over the short term, it should be noted that the review was limited to the comparison of only two studies [1,42].

In addition to the observation that composites showed poorer survival in this study, the same material and the duration of follow-up were seen to be significant moderators in the model estimating percentage survival of the restoration. This could be due to greater degradation of the material over time. However, although hybrid materials and ceramics should be regarded as the options of choice in indirect partial restorations in the posterior region due to their superior clinical performance, the use of composite as restoration material would be justified on the basis of its cost-benefit ratio, since the survival rates are still high despite the comparatively poorer performance with respect to other materials.

On analyzing the reasons for restoration failure, fracture (4%) was seen to be the most important cause of failure. Previous studies already identified fracture as the most common cause of failure [21,42]. This could be due to the fact that most of the analyzed studies used ceramic materials, which are known to be particularly prone to fracture.

The influence of the cementing technique has not been analyzed, due to the great heterogeneity of the adhesion protocols used. A large number of different types of adhesives and cements were employed, and in addition, the surface treatments according to the material involved and the isolation techniques were highly varied (Table 5). It should be noted that many articles failed to specify the cementing and adhesion materials used. Furthermore, analysis of the influence and behavior of the different adhesion protocols used for the indirect partial restorations in the posterior region was often made difficult by the lack of relevant data.

In order to assess the complications and clinical behavior of the restorations over time, most of the reviewed studies used criteria such as the modified USPHS or CDA, for example. These criteria have also been used in other reviews and meta-analyses [8,23,26,36]. It should be mentioned that although such criteria seek to standardize the evaluation of dental restorations, not all the studies made use of them for evaluation purposes. Furthermore, none of the publications analyzed the behavior of all the materials cited in the literature for onlay restorations in the posterior region. Most of the articles focused on complications related to a single material, or compared different treatment options, such as onlays, inlays, and crowns.

On analyzing the complications according to the criteria of the modified USPHS, 89.8% (95%CI 87.5–92.1) of the restorations were seen to correspond to category Alpha, while 9.8% (95%CI 7.7–1.9) corresponded to category Bravo. In contrast, the proportion of restorations classified as pertaining to categories Charlie or Delta was 0.1% (*p* = 0.855; *p* = 1). Based on the results obtained, it can be considered that all the restorations were regarded as acceptable. The most frequent complications were referred to changes in surface texture or color. Such complications could be attributed to degradation of the restoration material over time [36].

On analyzing the complications according to the criteria of the CDA system, 77.6% (95%CI 73.6–81.8) of the restorations were seen to correspond to success, i.e., the restorations were considered to meet the highest quality standards. In turn, 19.4% (95%CI 16.5–22.4) of the restorations were classified as corresponding to survival, i.e., the restorations suffered deterioration, though without requiring replacement. Lastly, 0.79% (95%CI 0.28–1.30) of the restorations corresponded to failure, i.e., they required replacement. The complications with the poorest scores were referred to the structure of the restoration (anatomy, surface texture and marginal integrity).

Independently of the evaluation system used, the classifications referred to clinical behavior and possible complications were seen to be similar in both cases.

Partial restorations, such as those analyzed in this review of onlays, should be regarded as the treatment of choice for the restoration of teeth in the posterior region, since they are predictable, exhibit good biomechanical behavior, and are conservative with the remaining dental tissues. Knowing the different materials available to make these restorations allows us to individualize the choice with criteria. However, at present, and given the rapidly evolving innovations in materials and adhesion techniques, it is not possible to define a concrete material as the clear gold standard—the choice having to be made on an individualized basis in each case. In turn, the lack of homogeneity among the published studies makes it difficult to establish objective comparisons among the different prosthodontic materials used. However, based on the results obtained in this research ceramics (ceramic reinforced with lithium disilicate better than conventional) are presented as the most reliable alternative in the long term. In addition, novel hybrid materials also exhibit good clinical behavior. Although the composite has less survival and greater degradation over time, each case should be analyzed individually, as it could be a good alternative economically.

Many in vitro studies have examined the behavior and durability of indirect restorations [23,71,72]. Although a number of clinical studies have analyzed the different materials used for onlay restorations over the long term, such materials have been studied separately [23,73,74]. Only a few articles have compared different materials in one same study—fundamentally ceramic and composite materials [21,45]. Furthermore, there are variations in the assessment criteria used in the different publications. Direct comparison of the results is therefore practically impossible. It also should be noted that because of the low quality of the studies, the conclusions drawn must be interpreted with caution. It would be advisable to carry out studies with larger sample sizes, evaluating all the materials proposed in the literature. On the other hand, use should be made of a more standardized methodology, with a presentation of results allowing for the comparison of the different studies in order to more precisely establish the behavior of these restorations. In this regard, additional long-term clinical studies are needed to examine the influence of the material used upon the clinical behavior of partial restorations in the posterior region.

## 5. Conclusions

Treatment in the form of indirect partial restorations is to be regarded as the option of choice in the posterior region, due to their good clinical performance and durability.

The performance of composites in terms of survival is significantly poorer than that of the hybrid materials or ceramics (*p* = 0.003), and composite materials are moreover also significantly affected by the passing of time (*p* < 0.001).

Fractures are the most common cause of restoration failure, while the most frequent causes of deterioration are related to the structure of the restoration (anatomy, color, marginal integrity, and surface texture).

## Figures and Tables

**Figure 1 ijerph-17-07582-f001:**
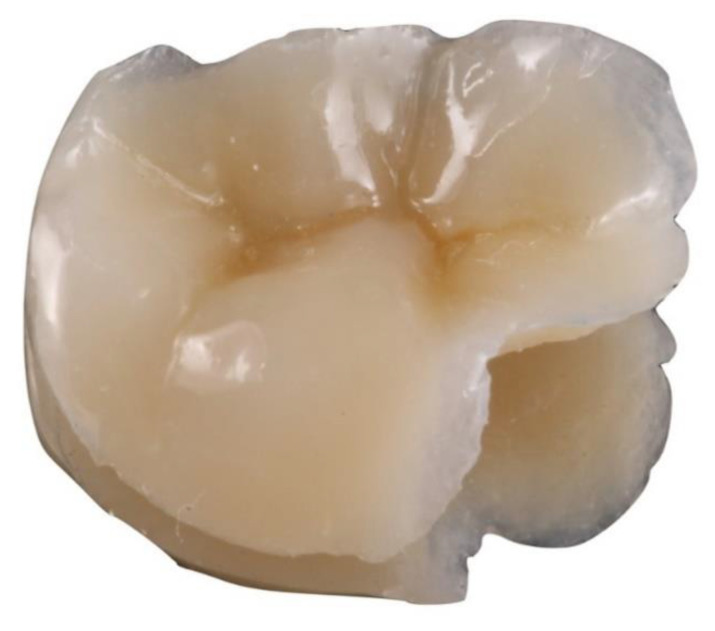
Partial restoration (onlay).

**Figure 2 ijerph-17-07582-f002:**
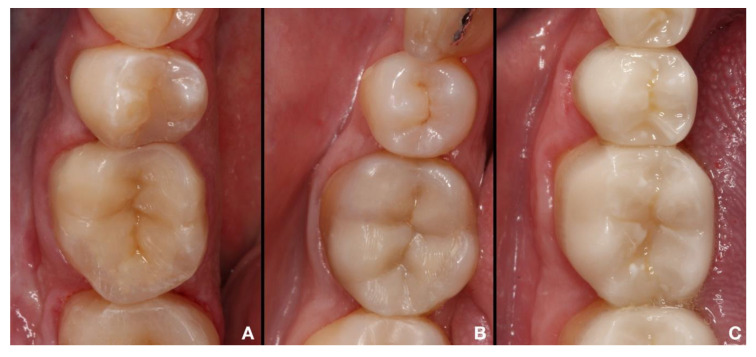
Indirect partial restorations made with different materials. (**A**). Composite. (**B**). Hybrid material. (**C**). Ceramics.

**Figure 3 ijerph-17-07582-f003:**
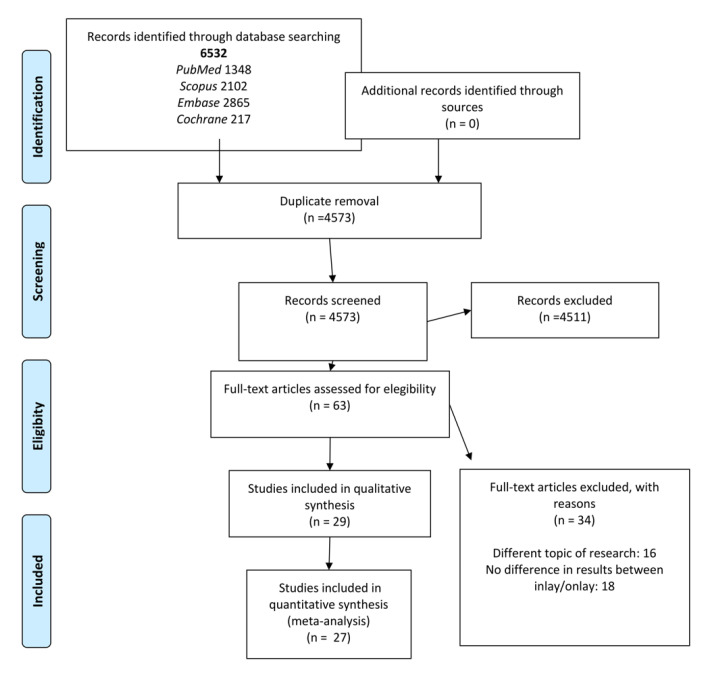
Preferred reporting items for systematic reviews and meta-analyses (PRISMA) flowchart.

**Figure 4 ijerph-17-07582-f004:**
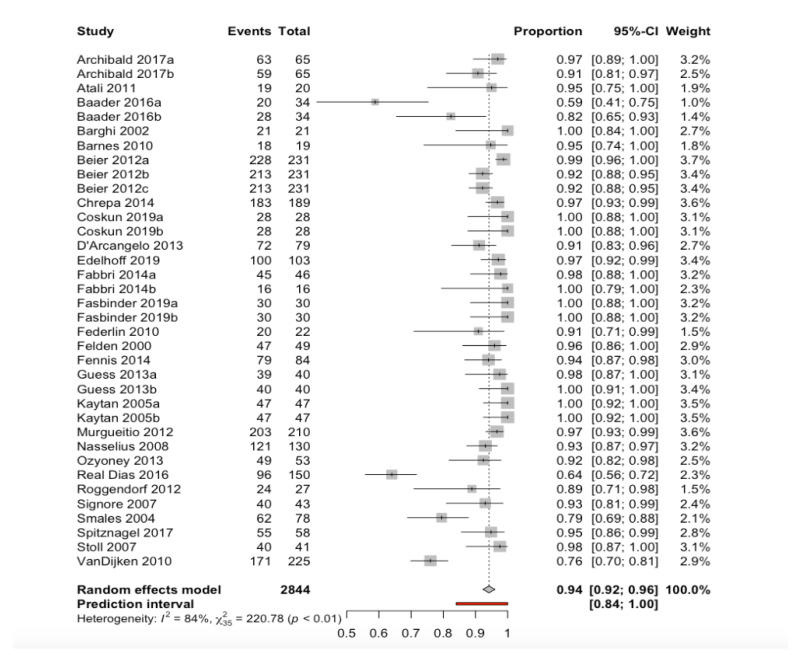
Forest plot indicating estimated percentage survival of the restorations.

**Figure 5 ijerph-17-07582-f005:**
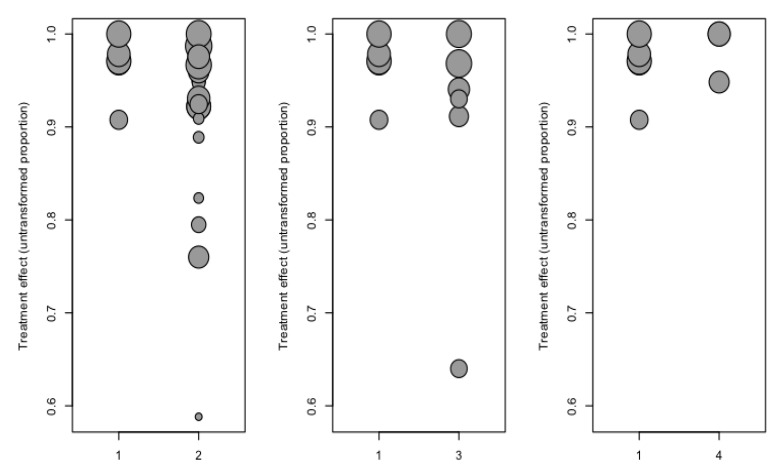
Scatter plot of the percentage survival regression analysis considering the covariates follow-up time and material. Categories: 1 = Disilicate; 2 = Feldspathic ceramic; 3 = Composites; 4 = Hybrids.

**Figure 6 ijerph-17-07582-f006:**
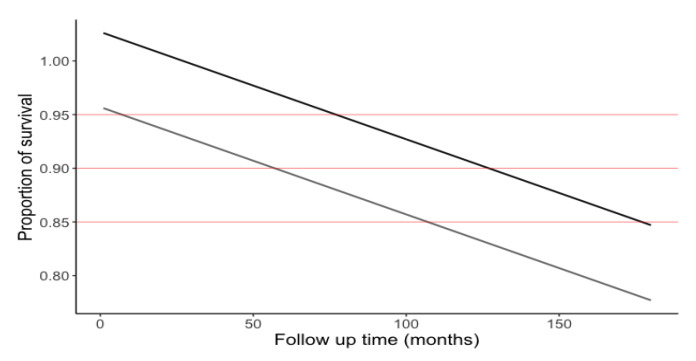
Restoration survival over time. The black line corresponds to survival over time of the restoration using ceramic or hybrid material, while the grey line corresponds to the significantly lesser survival of the composite restorations. It can be estimated that at 150 months of follow-up, percentage survival would be 88% for ceramic or hybrid materials versus 80% in the case of composite.

**Figure 7 ijerph-17-07582-f007:**
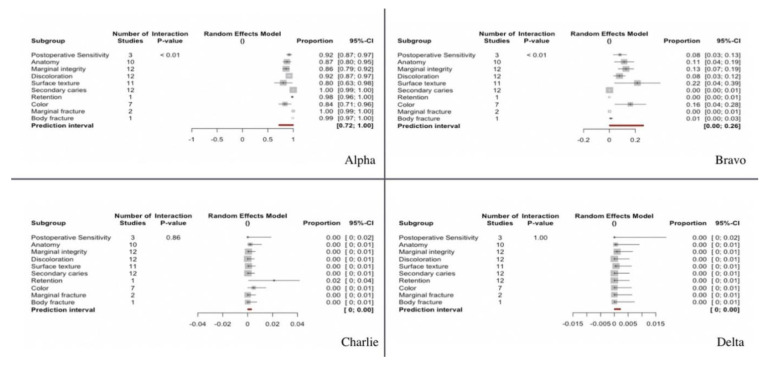
Forest plot according to clinical parameters for modified USPHS criteria category. Alpha, Bravo, Charlie and Delta.

**Figure 8 ijerph-17-07582-f008:**
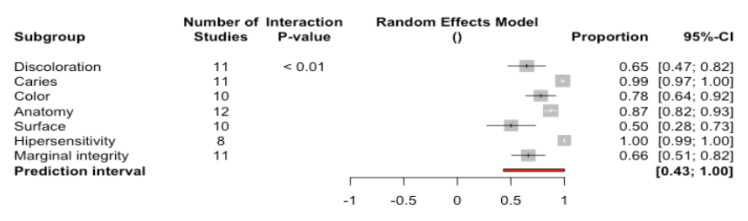
Forest plot according to parameters used to assess success.

**Figure 9 ijerph-17-07582-f009:**
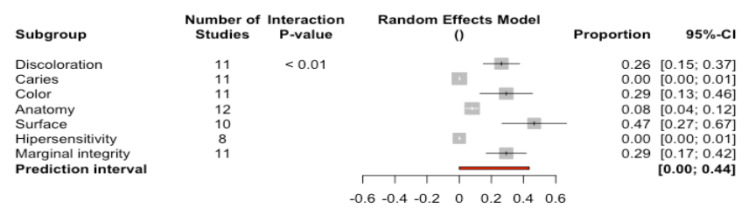
Forest plot according to parameters used to assess survival.

**Figure 10 ijerph-17-07582-f010:**
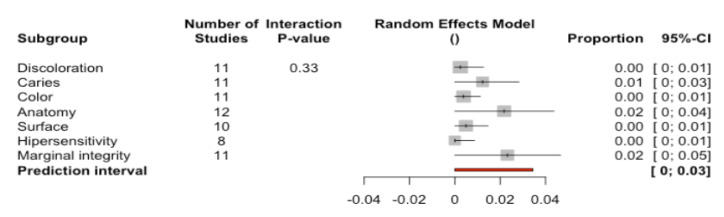
Forest plot according to parameters used to assess failure.

**Figure 11 ijerph-17-07582-f011:**
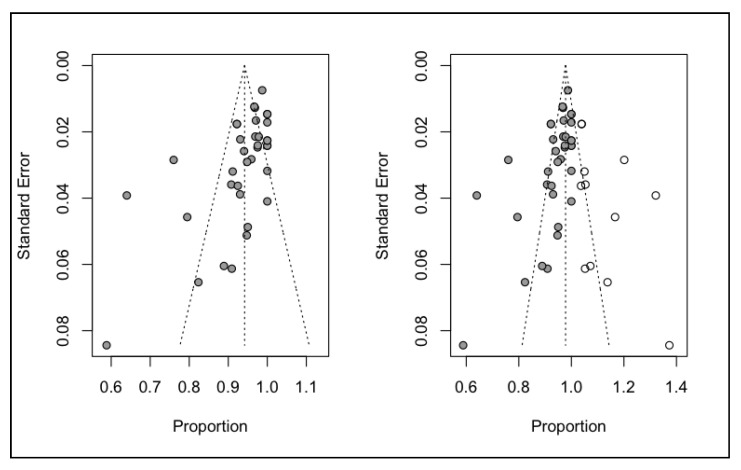
Funnel plot adjusted and unadjusted according to the Trim and Fill method for percentage survival.

**Table 1 ijerph-17-07582-t001:** Methodological quality of the articles according to the Newcastle–Ottawa scale for cohort studies.

AUTHOR (Year)	SELECTION				COMPARABILITY	OUTCOMES			TOTAL
	1	2	3	4	5–6	7	8	9	
Kaytan et al. (2005) [45]	*	*	* (Ceramic/composite)	*	*	*	*	*	8/9
Schulte et al. (2005) [46]	*	NA	-	*	-	*	*	*	5/9
Signore et al. (2007) [47]	*	NA	-	*	-	*	*	*	5/9
Beier et al. (2012) [48]	*	NA	* (bruxism yes/no)	*	*	*	*	*	7/9
Murgueitio et al. (2012) [49]	*	NA	-	*	-	*	*	*	5/9
Guess et al. (2013) [11]	*	*	* (pressed/CAD-CAM)	*	*	*	*	*	8/9
D’Arcangelo et al. (2014) [50]	*	NA	-	*	-	*	*	*	5/9
Fabbri et al. (2014) [51]	*	*	* (veneered/monolithic)	*	*	*	*	*	8/9
Chrepa et al. (2014) [52]	*	NA	-	*	-	*	*	*	5/9
Belli et al. (2016) [53]	*	*	* (emax CAD/empress CAD)	*	*	*	*	*	8/9
Baader et al. (2016) [54]	*	*	* (selective etching yes/no)	*	*	*	*	*	8/9
Archibald et al. (2017) [55]	*	*	* (emax press/emax CAD)	*	*	*	*	*	8/9

*NA: non-applicable. Scoring criteria: (1) Representativeness of the exposed cohort: truly representative (*) or somewhat representative (*); (2) Selection of the non-exposed cohort: drawn from the same community as the exposed cohort (*); (3) Ascertainment of exposure: secure record (e.g., surgical record) (*) or structured interview (*); (4) Outcome of interest not present at start of study: yes (*); (5–6) Comparability of cohorts based on design or analysis controlled for confounders: for the most important factor (*), for other factors (*); (7) Assessment of outcome: independent blind assessment or record linkage (*); (8) Follow-up long enough for outcomes to occur (6 months) (*); (9) Adequacy of follow-up of cohorts: subjects lost to follow-up unlikely to introduce bias–number lost ≤ 20% (*).

**Table 2 ijerph-17-07582-t002:** Methodological quality of the articles according to the PEDro scale for clinical trials.

AUTHOR (Year)	Criteria											Total
	1	2	3	4	5	6	7	8	9	10	11	
Felden et al. (2000) [56]	No	No	No	No	No	No	No	Yes	Yes	Yes	Yes	4
Barghi et al. (2002) [57]	Yes	No	No	No	No	No	No	Yes	Yes	Yes	Yes	5
Smales et al. (2004) [58]	No	No	No	No	No	No	No	Yes	Yes	Yes	Yes	4
Stoll et al. (2007) [59]	No	No	No	No	No	No	No	Yes	Yes	Yes	Yes	4
Naeselius et al. (2008) [60]	Yes	No	No	No	No	No	Yes	Yes	Yes	Yes	Yes	6
Federlin et al. (2010) [61]	Yes	Yes	No	Yes	No	No	Yes	Yes	Yes	Yes	Yes	8
Van Djken et al. (2010) [62]	Yes	No	No	No	No	No	No	Yes	Yes	Yes	Yes	5
Barnes et al. (2010) [63]	Yes	No	No	No	No	No	No	Yes	Yes	Yes	Yes	5
Atali et al. (2011) [64]	Yes	Yes	No	Yes	Yes	Yes	Yes	Yes	Yes	Yes	Yes	10
Roggendorf et al. (2012) [65]	Yes	No	No	No	No	No	No	Yes	Yes	Yes	Yes	5
Ozyoney et al. (2013) [66]	Yes	No	No	No	No	No	No	Yes	Yes	Yes	Yes	5
Fennis et al. (2014) [67]	Yes	Yes	No	Yes	No	No	No	Yes	Yes	Yes	Yes	7
Real Dias et al. (2016) [68]	Yes	No	No	No	No	No	No	Yes	Yes	Yes	Yes	5
Spitznagel et al. (2017) [29]	Yes	No	No	No	No	No	No	Yes	Yes	Yes	Yes	5
Cosçkun et al. (2019) [9]	Yes	Yes	No	Yes	No	No	No	Yes	Yes	Yes	Yes	7
Fasbinder et al. (2019) [69]	Yes	Yes	No	Yes	No	No	Yes	Yes	Yes	Yes	Yes	8
Edelhoff et al. (2019) [7]	Yes	No	No	No	No	No	No	Yes	Yes	Yes	Yes	5

Criteria: (1) The eligibility criteria were indicated; (2) The subjects were randomized to groups; (3) Allocation was concealed; (4) The groups were similar at baseline in terms of the most important prognostic indicators; (5) All subjects were blinded; (6) All those who administered the therapy were blinded; (7) All evaluators who measured at least one key outcome were blinded; (8) Measures of at least one key outcome were obtained for > 85% of the subjects initially allocated to groups; (9) All subjects for whom outcome measures were available underwent the treatment or control condition as allocated or, alternatively, data for at least one key outcome were analyzed on an intention-to-treat (ITT) basis; (10) Results of between-group statistical comparisons are reported for at least one key outcome; (11) The study provides both point measures and measures of variability corresponding to at least one key outcome.

**Table 3 ijerph-17-07582-t003:** Forest plot of the meta-analysis according to restoration material subgroups.

Subgroup	Number of Studies	Interaction *p*-Value	Proportion	95%CI
Lithium disilicate	8	<0.01	0.98	[0.96; 1.00]
Feldspathic ceramic	18		0.93	[0.90; 0.96]
Composites	5		0.90	[0.83; 0.98]
Hybrids	3		0.99	[0.96; 1.00]
**Prediction interval**				**[0.84; 1.00]**

**Table 4 ijerph-17-07582-t004:** Meta-regression analysis of percentage survival with duration of follow-up and restoration material as moderators. Reference material: disilicate.

Covariate	Beta Coefficient	95%CI	Z-Value	*p*-Value
Intercept	1.027	0.979, 1.075	42.3	<0.001
Follow-up time	−0.001	−0.002, −0.001	−3.49	0.001 *
Material: Feldspathic ceramic	−0.022	−0.069, 0.025	−0.93	0.353
Material: Composites	−0.064	−0.121, −0.007	−2.19	0.028 *
Material: Hybrids	−0.003	−0.073, 0.067	−0.08	0.940

Z-Value is the Z-score of the Z-test. *p*-Value is the level of marginal significance within the Z-test. * *p*-Value < 0.05 is statistically significant.

**Table 5 ijerph-17-07582-t005:** Analysis of articles included in the systematic review.

Author, Year	Title, Journal	Material	Luting Agent in Cementation
Fabbri et al. (2014) [51]	Clinical evaluation of 860 anterior and posterior lithium disilicate restorations: Retrospective study with a mean follow-up of 3 years and a maximum observational period of 6 years *The International Journal of Periodontics & Restorative Dentistry*	Feldspathic ceramic reinforced with lithium disilicate	**Restoration:** HF 20 sec. 4.5% (IPS ceramic gel) Monobond S. Optibond FL **Tooth:** Orthophosphoric ac. 37% 30 sec. Optibond FL Fluid composite resin (Gradia Direct Flow; Tetric EvoFlow) or dual cure composite systems (Variolink II)
Federlin et al. (2010) [61]	Controlled, prospective clinical split-mouth study of cast gold vs. ceramic partial crowns: 5.5 year results *American Journal of Dentistry*	Conventional feldspathic ceramic CAD-CAM (Vita 3D Master CEREC Mark II)	Dual cure composite cement (Variolink IIg/high viscosity)
D’Arcangelo et al. (2014) [50]	Five-year retrospective clinical study of indirect composite restorations luted with a light-cured composite in posterior teeth *Clin Oral Invest*	Composite	**Restoration:** EnaBond light-curing Aluminum oxide powder 50 μm **Tooth:** Immediate dentin sealing Orthophosphoric ac. 37% 30 sec. EnaBond light-curing EnaHeat Pre-heated (55 °C) photopolymerizing composite
Belli et al. (2016) [53]	Fracture Rates and Lifetime Estimations of CAD/CAM All-ceramic Restorations *Journal of Dental Research*	Feldspathic ceramic reinforced with lithium disilicate (emax CAD)/Leucite-reinforced ceramic (Empres CAD)	NR
Murgueitio et al. (2012) [49]	Three-Year Clinical Follow-Up of Posterior Teeth Restored with Leucite-Reinforced IPS Empress Onlays and Partial Veneer Crowns *American College of Prosthodontists*	Leucite-reinforced ceramic (IPS Empress)	**Restoration:** HF 20 sec. 5% (Ivoclar Vivadent) Monobond S. Excite DSC. **Tooth:** Orthophosphoric ac. 37% 20 sec. (selective enamel etching) Excite DSC. Dual cure resin cement (Variolink II)
Chrepa et al. (2014) [52]	The survival of indirect composite resin onlays for the restoration of root filled teeth: a retrospective medium-term study *International Endodontic Journal*	Composite (Gradia GC)	Dual cure, self-etching resin cement TotalCem
Archibald et al. (2017) [55]	Retrospective clinical evaluation of ceramic onlays placed by dental students *The Journal of Prosthetic Dentistry*	Feldspathic ceramic reinforced with lithium disilicate IPS emax Press/IPS emax CAD	**Restoration:** HF 20 sec. 10% (Prosthetic Etchant Gel) Monobond S. Multilink Primer or Excite DSC/or Scotchbond Universal Adhesive **Tooth:** Orthophosphoric ac. 35% 30 sec. Multilink Primer or Excite DSC/or Scotchbond Universal Adhesive Dual polymerizing cement (Variolink II or RelyX Ultimate)
Fennis et al. (2014) [67]	Randomized Control Trial of Composite Cuspal Restorations: Five-year Results *Journal of Dental Research*	Composite (Essentia, Kuraray)	**Restoration:** Blasting 15 sec. with aluminum oxide 50 μm Orthophosphoric ac. 37% Clearfil SE Bond primer mixed with Clearfil bond Activator **Tooth:** Orthophosphoric ac. 37% 20 sec. in enamel ED primer (self-etching primer) applied to enamel and dentin 60 sec. Dual cure composite resin (Panavia F)
Schulte et al. (2005) [46]	Longevity of ceramic inlays and onlays luted with a solely light-curing composite resin *Journal of Dentistry*	Leucite-reinforced ceramic (IPS Empress)	**Restoration:** HF (Vita ceramics etch) Monobond S. Heliobond. **Tooth:** Orthophosphoric ac. 37% Syntac classic Heliobond Photopolymerizing composite resin (Tetric)
Spitznagel et al. (2018) [29]	Polymer-infiltrated ceramic CAD/CAM inlays and partial coverage restorations: 3-year results of a prospective clinical study over 5 years *Clinical Oral Investigations*	Hybrid ceramic material (Vita Enamic CAD-CAM)	**Restoration:** HF 4.9% (IPS ceramic gel) Monobond S. Optibond FL **Tooth:** Orthophosphoric ac. 37% enamel 40 sec./dentin 15 sec. Syntac Primer Sytac Adhesive Heliobond Dual cure resin cement (Variolink II)
Guess et al. (2013) [11]	Prospective Clinical Split-Mouth Study of Pressed and CAD/CAM All-Ceramic Partial-Coverage Restorations: 7-Year Results *International Journal of Prosthodontics*	Feldspathic ceramic reinforced with lithium disilicate (IPS emax Press)/Leucite-reinforced ceramic (ProCAD) CAD-CAM	Photopolymerizing hybrid resin cement (Tetric/Syntac Classic)
Roggendorf et al. (2012) [65]	Seven-year clinical performance of CEREC-2 all-ceramic CAD/CAM restorations placed within deeply destroyed teeth *Clinical Oral Ivestigation*	Conventional feldspathic ceramic (VITABLOCS Mark II for CEREC)/Leucite-reinforced ceramic (ProCAD)	**Restoration**: HF 4.9% (IPS ceramic gel) Monobond S. Optibond FL **Tooth**: Orthophosphoric ac. 35% Sytac Photopolymerizing hybrid resin cement (Tetric Ceram) dual cure (Variolink Ultra)
Stoll et al. (2007) [59]	Survival of Inlays and Partial Crowns Made of IPS Empress After a 10-year Observation Period and in Relation to Various Treatment Parameters *Operative Dentistry*	Leucite-reinforced ceramic (IPS Empress)	**Restoration**: HF 4.9% (IPS ceramic gel) Monobond S. Optibond FL **Tooth**: Orthophosphoric ac. 35% Sytac Resin cement (Variolink cement or Variolink Ultra)
Beier et al. (2012) [48]	Clinical Performance of All-Ceramic Inlay and Onlay Restorations in Posterior Teeth *The International Journal of Prosthodontics*	Conventional feldspathic ceramic, sintered	Optibond FL Syntac Classic Optec cement 3M Cement Dual Zement Variolink High Viscosity (Dual cure composite cements)
Signore et al. (2008) [47]	A 4- to 6-Year Retrospective Clinical Study of Cracked Teeth Restored with Bonded Indirect Resin Composite Onlays *Int J Prosthodont*	Composite (Sculpture and Sculpture Plus)	**Restoration**: Monobond S. **Tooth**: Orthophosphoric ac. 15 sec. Ecusit PrimerMono (dentin adhesive) Orthophosphoric ac. 37% (total etch) Ecusit PrimerMono Dual resin cement (Variolink cement)
Real Dias et al. (2016) [68]	Prognosis of Indirect Composite Resin Cuspal Coverage on Endodontically Treated Premolars and Molars: An In Vivo Prospective Study *Journal of Prosthodontics*	Composite (Adoro System)	Cement RelyX Unicem-Tr
Felden et al. (2000) [56]	Retrospective clinical study and survival analysis on partial ceramic crowns: results up to 7 years. *Clin Oral Investig.*	Feldspathic ceramic reinforced with lithium disilicate (IPS Empress II)	Composite cement (dual cure, photopolymerizing cure)
Van Dijken et al. (2010) [62]	A prospective 15-year evaluation of extensive dentin-enamel-bonded pressed ceramic coverages. *Dent Mater.*	Leucite-reinforced ceramic, pressed	**Restoration**: HF 9.5% 2–3 sec. Orthophosphoric ac. 36% 20 sec. Monobond S **Tooth**: Orthophosphoric ac. 36% (enamel 10 sec., enamel and dentin 5 sec.) Composite cement (dual cure, photopolymerizing cure)
Barghi et al. (2002) [57]	Clinical evaluation of etched porcelain onlays: a 4-year report. *Compend Contin Educ Dent.*	Conventional feldspathic ceramic, sintered	Dual cure composite cement
Smales et al. (2004) [58]	Survival of ceramic onlays placed with and without metal reinforcement. *J Prosthet Dent.*	Conventional feldspathic ceramic, sintered (Mirage)	Dual cure composite cement (Mirage and Ultra-bond)
Kaytan et al. (2005) [45]	Clinical evaluation of indirect resin composite and ceramic onlays over a 24-month period. *Gen Dent.*	Leucite-reinforced ceramic, pressed	Dual cure composite cement
Naeselius et al. (2008) [60]	Clinical evaluation of all-ceramic onlays: a 4-year retrospective study. *Gen Dent.*	Leucite-reinforced ceramic, pressed	Dual cure and photopolymerizing cure composite cement
Barnes et al. (2010) [63]	Clinical evaluation of an all-ceramic restorative system: a 36-month clinical evaluation. *Am J Dent.*	Leucite-reinforced ceramic, pressed (Finesse All-Ceramic) with an ultra-low fusing porcelain (Finesse)	Dual cure composite cements (Esthetic resin cements, Enforce & Calibra)
Atali et al. (2011) [64]	IPS Empress onlays luted with two dual-cured resin cements for endodontically treated teeth: a 3-year clinical evaluation. *Int J Prosthodont*	Leucite-reinforced ceramic, pressed	Dual cure composite cements (Maxcem or Clearfil Esthetic Cement and DC Bond Kit luting systems)
Ozyoney et al. (2013) [66]	The efficacy of glass-ceramic onlays in the restoration of morphologically compromised and endodontically treated molars. *Int J Prosthodont.*	Feldspathic ceramic reinforced with lithium disilicate (IPS Empress II)	**Restoration**: HF 5% (IPS Empress ceramic etch) **Tooth**: Orthophosphoric ac. 35% Dentin bonding system: Solobond Plus Primer and Adhesive Dual cure high-viscosity composite cement (Bifix)
Baader et al. (2016) [54]	Self-adhesive Luting of Partial Ceramic Crowns: Selective Enamel Etching Leads to Higher Survival after 6.5 Years In Vivo. *J Adhes. Dent.*	Conventional feldspathic ceramic CAD-CAM (Vita Mark II)	**Restoration**: HF 5% (HF Vita ceramics etch) Monobond S (silano) **Tooth**:Orthophosphoric ac. 37% Auto-cure composite cement and auto-cure cement with selective etching (RelyX Unicem)
Edelhoff et al. (2019) [7]	Clinical performance of occlusal onlays made of lithium disilicate ceramic in patients with severe tooth wear up to 11 years. *Dental Materials.*	Feldspathic ceramic reinforced with lithium disilicate (IPS emax Press)	Syntac Total etch & rinse technique, Variolink II Professional Set, low viscosity, light-curing
Coskun et al. (2020) [9]	Evaluation of two different CAD-CAM inlay-onlays in a split-mouth study: 2-year clinical follow-up *J Esthet Restor Dent.*	Feldspathic ceramic reinforced with lithium disilicate (IPS emax CAD)/Hybrid ceramic material (Cerasmart)	**Restoration (Cerasmart):** Internal surface etched with 5% hydrofluoric acid (IPS Etching gel) 60 sec. **Restoration (IPS emax CAD):** Internal surface etched with 5% hydrofluoric acid (IPS Etching gel) 20 sec. Rinsed and silanized with Monobond Plus + Unfilled resin (Adhese Universal). **Tooth**: phosphoric acid gel 37% (Total Etch). Enamel 30 sec./dentin 15 sec. Adhese Universal 20 sec. Resin cement (Variolink Esthetic)
Fasbinder et al. (2020) [69]	Clinical evaluation of chairside Computer Assisted Design/Computer Assisted Machining nanoceramic restorations: Five-year status *J Esthet Restor Dent.*	Leucite reinforced ceramic (IPS Empress CAD)/Hybrid resin nanoceramic material (Lava Ultimate)	**Restoration (IPS Empress CAD):** 4.9% hydrofluoric acid gel 60 sec. Monobond Plus Scotchbond Universal Adhesive (3M) **Restoration (Lava Ultimate):** lightly air abraded with 30-μm silica (CoJet Sand; 3M) Scotchbond Universal Adhesive (3M) **Tooth (Variolink II cement):** phosphoric acid 37% 20 sec. Excite (Ivoclar) dentin bonding agent **Tooth (RelyX Ultimate cement):** Scotchbond Universal Adhesive Cement Variolink II and RelyX Ultimate

Author, year, title, journal, material and luting agent in cementation (distinguishing restoration and tooth preparation) were analyzed.

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
