# Peer review of "Clinical Behavior of Ceramic, Hybrid and Composite Onlays. A Systematic Review and Meta-Analysis"

_ijerph, 2020, doi:10.3390/ijerph17207582_

Round 1
Reviewer 1 Report
This manuscript describes the results of a systematic review on a well-known topic in restorative dentistry.
While reading the introduction, I was wondering about the scientific background which should be presented in this section. No detailed information is provided about state of the art direct/indirect restorations (regarding materials, dimensions). As is seems, this review is about indirect composite onlays. Therefore, the title is far away from beeing concise.
Because of this, I stop the detailed review of this manuscript at this point, as there are too many inconsistencies.
Author Response
Response to Reviewer 1
Comments and Suggestions for Authors
- ”This manuscript describes the results of a systematic review on a well-known topic in restorative dentistry.”
Dear Reviewer, Thank you for your comment but we believe that it can be very interesting for dentist readers, since under our knowledge and although there are many articles in the literature that analyze the behavior of onlays, there is no work in literature that analyzes all the materials for onlays, including recently available CAD-CAM hybrid materials, among others. As well as their behavior based on each of these materials and distinguishing them. In addition, many of the research does not distinguish the results obtained depending on whether the restoration is of type onlay or inlay. Which we consider important given that inlays tend to have superior survival because of their smaller size and lower functional involvement. We consider that our work provides a very interesting update for dentists, on a type of restoration that is booming in daily clinical practice. This investigation can help clinical decision-making (on the material of choice, possible complications, survival rates, etc.), because to this day there is no gold standard option in such treatments.
- ”While reading the introduction, I was wondering about the scientific background which should be presented in this section. No detailed information is provided about state of the art direct/indirect restorations (regarding materials, dimensions). As is seems, this review is about indirect composite onlays. Therefore, the title is far away from beeing concise.”
Dear Reviewer, Thank you for your comment. The text has been revised introducing the detailed explanation as you advise. We have revised the introduction, restructured it and clarified the different concepts addressed in the paper.
“….Partial restorations are an alternative to conventional crowns, in view of the growing demand for minimally invasive restorations, since crowns or complete covering restorations imply important loss of tooth structure, with macromechanical and more destructive preparation invasive preparation of the dental tissues [7,9–13]. In this regard, onlay partial restorations have become a conservative treatment option thanks to their good aesthetic outcomes, durability, resistance to wear, color stability, biocompatibility and high long-term survival rates [14–16]. The development of this treatment option has allowed lesser tooth reduction during preparation of the restoration, and thus greater preservation of the dental tissues. Partial restorations are classified according to the area to be restored as inlays (without covering the cusps), onlays (covering at least one cusp) [17], and overlays (covering all cusps) [18–21]. In addition to classification based on the area to be restored, two types of partial restorations have been defined according to the manufacturing method employed: direct and indirect restorations. In the former case In direct restorations, the restoring material (which is limited to composite) is placed directly in the defect or cavity, affording greater preservation of tooth structure. These restorations are mainly indicated in cases of lesser dental destruction [22–26]. The indirect restoration technique in turn involves preparation of the restoration outside the mouth (using ceramics, hybrids or composite). Compared with the direct technique, the indirect restoration approach affords cusp protection, with reinforcement of the compromised tooth [19,21,27,28]. Within addition to the evolution of adhesive techniques and materials, modern digital methods for manufacturing indirect restorations have been developed. Computer-aided design and computer-aided manufacturing (CAD-CAM) systems, along with the improvement of intraoral scanners, constitute alternative to the conventional method for manufacturing high-quality indirect restorations [29,30]. These novel approaches allow the restoration of lost tooth tissues in a single visit, with consequent reduction of the overall treatment time [31]. Nevertheless, the limitations of the different techniques (analog and digital) must be known in order to establish the workflow best suited to each case [32], since correct manipulation of the material and adequate selection of the manufacturing or adhesion technique are key factors influencing restoration success or failure [23]. The use of such restorations has increased as a result of advances in adhesion and cementing technologies, affording greater bonding between the tooth and the restoration material, and thus allowing for more conservative treatments as well as increased easiness of use [33–36]. Restoration materials have also been influenced by these advances, evolving from the use of materials such as gold or amalgam, with a long history of clinical success and biocompatibility [23], to more current materials such as ceramics (conventional feldspathic ceramic, leucite-reinforced ceramic, lithium disilicate ceramic), hybrid materials (resin nanoceramic and hybrid ceramic) or composite resins [37,38].
….”
As this review and meta-analysis was carried out to analyze the survival of onlay restorations in the posterior sector, examining the different materials used (such as feldspathic ceramic reinforced with lithium disilicate, conventional feldspathic ceramic or feldspathic ceramic reinforced with leucite, hybrid materials and composite) and identifying the types and frequencies of complications reported in randomized controlled trials (RCTs) and retrospective and prospective studies, the title has been modified, according to your suggestion, in order to avoid confusion.
“Clinical behavior of ceramic, hybrid and composite onlays. A systematic review and meta-analysis”.
- ”Because of this, I stop the detailed review of this manuscript at this point, as there are too many inconsistencies.”
Dear Reviewer, we hope that we have been able to solve the inconsistencies and be able to clarify your doubts, in order to make the work interesting to you and to your liking. We have rewritten the introduction, reordered the ideas and improved the document.

Reviewer 2 Report
English is very poor, which makes it difficult to read the text
The introduction is not written in a structured manner, the authors describe modern techniques and materials used for the manufacture of onlays in an extremely fragmentary and non-systematic manner. There is no mention of modern digital methods of manufacturing the indirect restorations, which allow restoring lost tooth tissues in one visit. Please rewrite.
Many professional terms are incorrect, for example, line 46 “destructive preparation” the established professional term is the “invasive preparation” , line 55 “imprint” the correct term is “impression” etc. Line 72-73 requires the complete rewriting, since it contains a large number of errors, both semantic and grammatical. Line 94 – “posterior sector” the correct term is “posterior region”
In the text, the authors contradict their own statements, for example, at lines 57-58, the authors indicated that the tabs have "wear compatible with that of the opposing natural dentition", while at lines 68-69 they postulate that "restorations are material wear or wear of the opposing teeth"
Line 157 - are the authors writing about failed implants? This article is about onlays, not implants.
The authors point out that they have analyzed 29 studies, while “Eight article analyzed feldspathic ceramic reinforced with lithium disilicate, while feldspathic ceramic (conventional or reinforced with leucite) was evaluated as restoration material in 18 articles. It is indicated below that in five more articles the composite was studied. 18 +8 +5 = 31 ???
Please check the manuscript for mistypes.
Please rewrite the discussion to make the idea of the manuscript clear.
I strongly recommend getting editing help from someone with full professional proficiency in English and to double-check all specific professional terms.
Author Response
Response to Reviewer 2
Comments and Suggestions for Authors
- “English is very poor, which makes it difficult to read the text”.
Dear Reviewer, Thank you for taking the time to read this article. We have forwarded the comments to the translator to improve the entire wording.
- “The introduction is not written in a structured manner, the authors describe modern techniques and materials used for the manufacture of onlays in an extremely fragmentary and non-systematic manner. There is no mention of modern digital methods of manufacturing the indirect restorations, which allow restoring lost tooth tissues in one visit. Please rewrite.”
Dear Reviewer, the paper has been modified and the introduction rewritten. Modern digital methods of manufacturing the indirect restorations have been mentioned.
“…Onlays Partial restorations are an alternative to conventional crowns, in view of the growing demand for minimally invasive restorations, since crowns or complete covering restorations imply important loss of tooth structure, with macromechanical and more destructive preparation invasive preparation of the dental tissues [7,9–13]. In this regard, onlay partial restorations have become a conservative treatment option thanks to their good aesthetic outcomes, durability, resistance to wear, color stability, biocompatibility and high long-term survival rates [14–16]. The development of this treatment option has allowed lesser tooth reduction during preparation of the restoration, and thus greater preservation of the dental tissues. Partial restorations are classified according to the area to be restored as inlays (without covering the cusps), onlays (covering at least one cusp) [17], and overlays (covering all cusps) [18–21]. In addition to classification based on the area to be restored, two types of partial restorations have been defined according to the manufacturing method employed: direct and indirect restorations. In the former case In direct restorations, the restoring material (which is limited to composite) is placed directly in the defect or cavity, affording greater preservation of tooth structure. These restorations are mainly indicated in cases of lesser dental destruction [22–26]. The indirect restoration technique in turn involves preparation of the restoration outside the mouth (using ceramics, hybrids or composite). Compared with the direct technique, the indirect restoration approach affords cusp protection, with reinforcement of the compromised tooth [19,21,27,28]. Within addition to the evolution of adhesive techniques and materials, modern digital methods for manufacturing indirect restorations have been developed. Computer-aided design and computer-aided manufacturing (CAD-CAM) systems, along with the improvement of intraoral scanners, constitute alternative to the conventional method for manufacturing high-quality indirect restorations [29,30]. These novel approaches allow the restoration of lost tooth tissues in a single visit, with consequent reduction of the overall treatment time [31]. Nevertheless, the limitations of the different techniques (analog and digital) must be known in order to establish the workflow best suited to each case [32], since correct manipulation of the material and adequate selection of the manufacturing or adhesion technique are key factors influencing restoration success or failure [23]. The use of such restorations has increased as a result of advances in adhesion and cementing technologies, affording greater bonding between the tooth and the restoration material, and thus allowing for more conservative treatments as well as increased easiness of use [33–36]. Restoration materials have also been influenced by these advances, evolving from the use of materials such as gold or amalgam, with a long history of clinical success and biocompatibility [23], to more current materials such as ceramics (conventional feldspathic ceramic, leucite-reinforced ceramic, lithium disilicate ceramic), hybrid materials (resin nanoceramic and hybrid ceramic) or composite resins [37,38].…”
- “Many professional terms are incorrect, for example, line 46 “destructive preparation” the established professional term is the “invasive preparation” , line 55 “imprint” the correct term is “impression” etc. Line 72-73 requires the complete rewriting, since it contains a large number of errors, both semantic and grammatical. Line 94 – “posterior sector” the correct term is “posterior region””.
Dear Reviewer, Thank you for your comment. The incorrect terms have been corrected, following your recommendation, and the article has been sent to the translator to improve the entire wording.
“As a professional native English-speaking medical translator, and upon request from the signing authors, I declare that I have fully revised the grammar and structure of the article entitled “Clinical behavior of ceramic, hybrid and composite onlays. A systematic review and meta-analysis”.
/
Ira (Joe) Perkins
ID no. 22588209R”.
“…with macromechanical and more destructive preparation invasive preparation of the dental tissues [7,17–21].”
Some sentences have been deleted as line 55 and 72-73, among others.
“The present systematic review and meta-analysis was carried out to analyze the survival of onlay restorations in the posterior region sector.”
- “In the text, the authors contradict their own statements, for example, at lines 57-58, the authors indicated that the tabs have "wear compatible with that of the opposing natural dentition", while at lines 68-69 they postulate that "restorations are material wear or wear of the opposing teeth"”.
Dear Reviewer, Thank you for your comment. Some sentences have been deleted with the aim of restructuring the introduction and improving it. But with regard to ceramics:
“Ceramic materials are fragile and more vulnerable to fracture than composite materials, though they are also harder than the latter and are therefore more resistant to wear. However, for this same reason they may induce wear of the surface of the opposing tooth [1,38].”
- “Line 157 - are the authors writing about failed implants? This article is about onlays, not implants.”
Thank you for your comment, the article has been modified accordingly because it was a traduction mistake.
“The CDA is a variation of the USPHS system. Both instruments are based on an ordinal scale, rating restorations as being either “acceptable” or “not acceptable”. “Success” refers to successful restorations; “survival” refers to restorations that are not intact but survive; and “failure” refers to failed implants restorations.”
- “The authors point out that they have analyzed 29 studies, while “Eight article analyzed feldspathic ceramic reinforced with lithium disilicate, while feldspathic ceramic (conventional or reinforced with leucite) was evaluated as restoration material in 18 articles. It is indicated below that in five more articles the composite was studied. 18 +8 +5 = 31 ???”
Dear Reviewer, Thank you for your comment. The numbers described in the article are correct, because:
“A total of 29 articles were entered in the qualitative analysis. The sample sizes ranged from 14-231 restorations, and the duration of follow-up varied between 24-180 months. The materials analyzed were: feldspathic ceramic reinforced with lithium disilicate, conventional feldspathic ceramic or feldspathic ceramic reinforced with leucite, hybrid materials and composite. Eight articles analyzed feldspathic ceramic reinforced with lithium disilicate, while feldspathic ceramic (conventional or reinforced with leucite) was evaluated as restoration material in 18 articles. Two of these studies compared both materials (ceramic reinforced with lithium disilicate and feldspathic ceramic reinforced with leucite). One of them compared two lithium disilicate ceramics, differentiating their laboratory manufacture (press/CAD), and another compared two different ceramics (conventional and reinforced with leucite). In addition, we analyzed three articles in which hybrid materials were used. This type of material was compared against ceramic reinforced with leucite in one article and versus ceramic reinforced with lithium disilicate in another. In five articles the restoration material considered was composite.”
Eight articles analyzed feldespathic ceramic reinforced with lithium disilicate of which in 3, two different materials are analyzed in the same work: ceramic reinforced with lithium disilicate + hybrids (Coskun 2019), and in 2 articles ceramic reinforced with lithium disilicate + ceramic reinforced with leucite (Guess 2013, Belli 2016).
While feldspathic ceramic (conventional or reinforced with leucite) was evaluated as restoration material in 18 articles. Two of these studies compared both materials (ceramic reinforced with lithium disilicate and feldspathic ceramic reinforced with leucite, Guess 2013, Belli 2016). One of them compared two lithium disilicate ceramics, differentiating their laboratory manufacture (press/CAD), and another compared two different ceramics (conventional and reinforced with leucite, Roggendorf 2012). Hybrid material was compared against ceramic reinforced with leucite in one article (Fasbinder 2019). (…)
As a summary: “Of the 29 articles analyzed, 5 of them evaluated and compared two different materials within the same research. In 5 articles only ceramic reinforced with lithium disilicate was analyzed, and in another three publications ceramic reinforced with lithium disilicate was compared with another material (overall, ceramic reinforced with lithium disilicate was used in 8 articles). In 13 articles only feldspathic ceramics (conventional or reinforced with leucite) were analyzed, and in another 5 publications feldspathic ceramics were compared with another material (overall, 18 articles used this material). Hybrid materials were analyzed in one article and in two others these materials were compared against another material (a total of three articles used hybrid materials). Composites were used in 5 investigations.”
- ”Please check the manuscript for mistypes.”
Dear Reviewer, Thank you for your comment. The manuscript has been checked for mistypes.
- ”Please rewrite the discussion to make the idea of the manuscript clear.”
Dear Reviewer, the entire manuscript has been checked to make the idea of the manuscript clear. We have rewritten and reordered introduction and discussion according to your suggestion, with the aim of clarifying concepts and providing you a clearer and more understandable structure for the reader.
- ”I strongly recommend getting editing help from someone with full professional proficiency in English and to double-check all specific professional terms.”
Dear Reviewer, Thank you for your recommendation. We have forwarded the comments to the translator to improve the entire paper.
“As a professional native English-speaking medical translator, and upon request from the signing authors, I declare that I have fully revised the grammar and structure of the article entitled “Clinical behavior of ceramic, hybrid and composite onlays. A systematic review and meta-analysis”.
/
Ira (Joe) Perkins
ID no. 22588209R”.

Round 2
Reviewer 2 Report
Dear authors, thank you for your work. You have significantly improved the quality of the text.
However, please 1) think about 1) shifting some Figures (as Figure 5. Forest plot… - Figure 8) to the supplementary file or making the collage; 2) strengthening the Discussion part by the explaining the novelty of the results obtained, and 3) adding some suggestions to the possible readers from the practical dentistry.
Author Response
Manuscript ID: ijerph-941579
Title: Clinical behaviour of ceramic, hybrid and composite onlays. A systematic review and meta-analysis.
Response to Reviewer 2
“Dear authors, thank you for your work. You have significantly improved the quality of the text.
However, please 1) think about 1) shifting some Figures (as Figure 5. Forest plot… - Figure 8) to the supplementary file or making the collage; 2) strengthening the Discussion part by the explaining the novelty of the results obtained, and 3) adding some suggestions to the possible readers from the practical dentistry.”
Dear Reviewer, Thank you for your comments and for your review.
- Figures 5-8 have been making the collage as you advice.
Figure 7. Forest plot according to clinical parameters for modified USPHS criteria category. Alpha, Bravo, Charlie and Delta.
- Discussion part have been strengthened.
With regard to the novelty of the results obtained:
“Most of the investigations and meta-analyses published to date have evaluated restorations of this kind centered on a single restoration material. Many of the published studies do not differentiate the results obtained according to the extent of the restoration (inlay or onlay). Furthermore, to the best of our knowledge, no studies have examined all the available materials for the manufacture of onlay restorations with the purpose of assessing their clinical behavior over time and of defining the gold standard or ideal material for the preparation of such restorations. In this regard, the present meta-analysis has examined the survival and possible complications of partial restorations in the sector posterior posterior region, including all materials currently available for the manufacture of onlays, among which are the novel hybrid CAD-CAM materials.”
- Some suggestions have been added for the possible readers for the practical dentistry.
We hope that the changes will be to your liking, thank you very much for your interest.
